# Principled Data Generation for MetaBBO via Active Task Selection

## Abstract

Meta-Black-Box Optimization (MetaBBO) aims to acquire generalized optimization strategies by training on extensive sets of functions. While traditional approaches predominantly rely on fixed benchmarks, recent efforts have shifted towards maximizing landscape diversity to broaden training distribution coverage. However, we identify a critical *Diversity-Quality Gap*: Simply increasing landscape variety often yields tasks that are misaligned with the agent's evolving capabilities. To address this, we propose *Hierarchical Active Task Selection (HATS)*, a principled framework that constructs an automated curriculum based on the agent's potential for improvement. HATS introduces a regret-based utility metric measuring the performance gap between the current policy and a competent baseline to guide a bi-level selection process. Specifically, it combines a multi-armed bandit for dynamic function class weighting with an active sampler for parameter replay. Experiments demonstrate that HATS significantly outperforms diversity-driven baselines and achieves superior generalization on real-world tasks with fewer training steps, highlighting the value of quality-based data generation in MetaBBO.

## 1. Introduction

Black-Box Optimization (BBO), which involves optimizing objective functions where analytical forms and gradient information are inaccessible, is a crucial challenge in real-world complex optimization problem, ranging from hyperparameter tuning (Arango et al., 2021), molecule discovery (Shin et al., 2025) and chip placement (Shi et al., 2023; Xue et al., 2025). Driven by the need to automate these processes, the field of BBO is undergoing a paradigm shift from manually designed heuristics to data-driven au-

tomation (Song et al., 2024; Ma et al., 2025d). Meta-Black-Box Optimization (MetaBBO) aims to learn generalized optimization strategies by training parameterized policies on extensive distributions of problem instances (Ma et al., 2023; 2025c). Unlike traditional BBO optimizers defined by static rules, MetaBBO agents distill search patterns from interaction from the training functions, making their performance heavily dependent on the quality and structure of the training distribution (Wang et al., 2026).

To prevent overfitting to static instances, current approaches in MetaBBO predominantly generate data via open-loop diversity maximization. Domain Randomization (DR) (Tobin et al., 2017), a technique of randomly sampling function instances, is often employed to augment the training set (Lange et al., 2023; Chen et al., 2024; Hansen et al., 2021). A recent pioneering work, Diverse-BBO (Wang et al., 2026), incorporates Exploratory Landscape Analysis (ELA) features (Mersmann et al., 2011) to generate a set of diverse functions. These approaches assume that a training set spanning a wide range of landscape features guarantees a robust learned policy. However, in this paper, we challenge this assumption. Through a quantitative analysis of instance quality, we identify a Diversity-Quality Gap that simply maximizing landscape diversity is misaligned with the agent's capabilities. This observation underscores the necessity of a principled data generation framework for MetaBBO that is aware of function instance's utility to the current agent.

To address this, we propose defining task utility by the agent's dynamic potential for improvement. Recognizing that global optima are often intractable in generated functions, we introduce a regret-based utility metric that measures the performance gap between the current MetaBBO policy and a competent baseline (e.g., MadDE (Biswas et al., 2021)). This metric serves as a precise learning signal, guiding the generator to construct a curriculum of tasks that are challenging yet solvable relative to the teacher. Using the utility metric, we reformulate the data generation framework for MetaBBO by scheduling instances based on their utilities, as shown in Figure 1. This framework views the generation of instances as a closed-loop process, thereby enabling automated curriculum learning for MetaBBO.

Practically, we instantiate this principle as Hierarchical Ac-

---

[1]Anonymous Institution, Anonymous City, Anonymous Region, Anonymous Country. Correspondence to: Anonymous Author <anon.email@domain.com>.

Preliminary work. Under review by the International Conference on Machine Learning (ICML). Do not distribute.

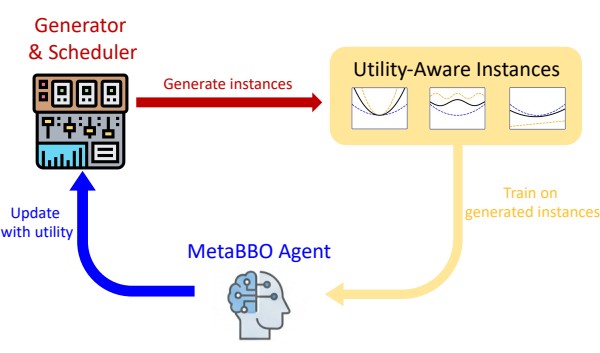

*Figure 1.* Illustration of our proposed utility-aware framework for MetaBBO, where the generator module interacts with the agent by generating utility-aware instances for agent training and receiving utility feedback from the agent.

tive Task Selection (HATS), a framework that leverages the inherent structure of BBO problems. HATS treats the instance space as a hierarchy $\mathcal{G} = \langle \mathcal{C}, \Theta \rangle$, separating global topology (e.g., funnel structures and multi-modality) from local geometric distortions (e.g., rotation and shifting) (Mersmann et al., 2011; Hansen et al., 2021). Specifically, the framework employs a bi-level active selection strategy: 1) A robust multi-armed bandit dynamically allocates the training budget across function classes to ensure topological coverage and prevent catastrophic forgetting; 2) An active sampler prioritizes distortion parameters that maximize the regret-based utility, ensuring the agent continuously trains on the most informative instances for its current stage (Jiang et al., 2021b;a).

Empirically, we examine the performance of HATS under SYMBOL (Chen et al., 2024), a representative MetaBBO method, on collected optimization problems from MetaBox (Ma et al., 2023; 2025c), a widely used benchmark for MetaBBO. Experimental results validate the effectiveness of HATS over diversity-based data generation methods. Extended analysis on function class distribution throughout training further demonstrates the learning dynamics of the MetaBBO agent.

Our contributions are summarized as follows:

- We empirically demonstrate the Diversity-Quality Gap in MetaBBO, showing that feature diversity correlates poorly with training efficiency.

- We propose HATS, a data generation framework that constructs an adaptive curriculum by actively selecting function class and parameters based on the performance gap towards a baseline.

- We validate HATS on unseen instances from benchmark suites (CEC and BBOB) and real-world applications including Protein Docking and UAV path planning, showing significant improvements in generalization over state-of-the-art diversity-driven methods.

## 2. Preliminaries and Related Works

### 2.1. BBO and MetaBBO

**BBO.** A standard BBO problem involves minimizing a black-box objective function $f : \Omega \mapsto \mathbb{R}$, where $\Omega \subseteq \mathbb{R}$ is the search space of dimension $D$. The goal is to find $\mathbf{x}^* = \arg\min_{\mathbf{x} \in \Omega} f(\mathbf{x})$. In BBO, the analytical form and gradient information of $f$ are inaccessible, thus a BBO optimizer $\mathcal{A}$ interacts with $f$ solely through queries, receiving a response $y = f(\mathbf{x})$ for each query $\mathbf{x}$, constrained by a maximum budget $T$ of function evaluations.

**MetaBBO.** Leveraging meta-learning techniques (Thrun & Pratt, 1998; Finn et al., 2017), MetaBBO aims to automate the design of BBO algorithms by learning a policy that generalizes across a distribution of problem instances (Ma et al., 2025d), showing superior applicability to real-world complex optimization problems (Lu et al., 2026). This process can typically be reformulated by a bi-level optimization problem:

- **Inner level**: The optimizer $\mathcal{A}$, controlled by a parameterized policy $\pi_\phi$ (where $\phi$ denotes the meta-parameters, e.g., weights of a neural network), executes the optimization process on a specific problem instance $f$. This generates a trajectory $\tau_f = \{(\mathbf{x}_t, y_t)\}_{t=1}^{T}$.

- **Meta level**: The objective of MetaBBO is to find the optimal meta-parameters $\phi^*$ that maximize the expected performance metric $\mathcal{M}$, e.g., negative regret, over a target problem distribution $p(f)$. Formally, it can be written by:

$$\phi^* = \arg\max_{\phi} \mathbb{E}_{f \sim p(f)}[\mathcal{M}(\mathcal{A}(\pi_\phi, f))].$$

Targeting at different parameterized perspectives, existing MetaBBO methods can be categorized into three lines (Ma et al., 2025d). The first category is algorithm selection, which maintains a pool of algorithms and fits a selector for optimal algorithms (Guo et al., 2024; Cenikj et al., 2024). The second type of MetaBBO is algorithm configuration (Hutter et al., 2009; Biedenkapp et al., 2020), which dynamically adjusts parameter values for a fixed optimizer (Xue et al., 2022; Lu et al., 2025; Guo et al., 2025a). Due to the efficiency of leveraging offline data (Ma et al., 2025a) and interpretablity (Chen et al., 2024), algorithm generation recently has emerged as a novel solution for MetaBBO.

### 2.2. Data Generation Issues in MetaBBO

A critical limitation in current MetaBBO research is the static nature of the training distribution $p_{\text{train}}(f)$. Existing works typically train the policies via a predefined training set, e.g., using fixed benchmark function suites like BBOB (Hansen et al., 2021) or static augmenta-

tions. Relying on a fixed set of functions leads to overfitting (Wichrowska et al., 2017), where the policy $\pi_\phi$ memorizes specific landscape features rather than learning generalizable strategies (Lange et al., 2023).

To mitigate this overfitting, recent works have tried to augment the training problem set to enrich the training data, thus to enhance the performance of the learned policy. The most prevalent approach is to apply Domain Randomization (DR) augmentation (Tobin et al., 2017), e.g., randomly generated shift and rotation, on a class of base functions (Chen et al., 2022; Lange et al., 2023; Chen et al., 2024; Song et al., 2025). To further formulate diversity for BBO functions, Wang et al. (2026) recently introduced Exploratory Landscape Analysis (ELA) features, a widely used tool for BBO instance analysis (Mersmann et al., 2011; Ma et al., 2025b), to measure instances' diversity and construct a set of diverse instances via genetic programming (Long et al., 2023). In Section 3, we further point out that simply augmenting diversity of training instances does not fit the need of MetaBBO agents. Notably, MetaBBO agents are often instantiated via Reinforcement Learning (RL). As pointed out by the RL community, in different training stages, RL agents may require environments with different difficulties (Parker-Holder et al., 2022; Portelas et al., 2020), which we will discuss in the next subsection.

### 2.3. Automated Curriculum Learning

Automated curriculum learning aims to enhance the generalization and sample efficiency of RL agents by dynamically adapting the training task distribution to the agent's evolving capabilities (Portelas et al., 2020). While DR (Tobin et al., 2017) serves as a foundational baseline for such generalization, it often suffers from sample inefficiency due to uniform sampling over the task space. To address this, Unsupervised Environment Design (UED) frames curriculum generation as a regret maximization game (Dennis et al., 2020). Jiang et al. (2021b) introduced Prioritized Level Replay (PLR), which selectively replays levels with high value loss, significantly improving the sample efficiency, and Wang et al. (2022) pointed out that the levels should be remained in a learnable zone. To extend PLR to stochastic environments, PLR$^\perp$ (Jiang et al., 2021a) was proposed to decrease noise from epistemic uncertainty, while ACCEL (Parker-Holder et al., 2022) integrates evolutionary strategies to progressively mutate and generate complex environments from the PLR buffer (Bhatt et al., 2022). Recently, these UED principles have been successfully adapted to more complex domains, such as generating synthetic environments for LLM agents (Guo et al., 2025b).

## 3. Measuring Data Efficiency for MetaBBO

In this section, we re-examine the data efficiency for MetaBBO. In Section 3.1, we challenge the common practice that maximizes instance feature diversity and find the poor correlation between instance diversity and quality. To efficiently quantify an instance's quality for an agent, we define performance gap-based utility in Section 3.2. In Section 3.3, we further discuss the instance structures of BBO functions, which motivates us to design a principled data generation pipeline for MetaBBO.

### 3.1. Does Higher Data Diversity Lead to Better Training Efficiency?

As discussed in Section 2.2, recent works prioritize maximizing the feature diversity of training set to prevent overfitting. These approaches operate under the implicit assumption that a training distribution with sufficient diversity, which spans a broad range of landscape characteristics, inherently guarantees a robust and generalizable MetaBBO policy (Wang et al., 2026). However, this raises a fundamental yet overlooked question: *Does maximizing feature diversity necessarily lead to better training efficiency for the MetaBBO agents?* We conduct an illustrative experiment to answer this question by analyzing the correlation between feature diversity and training efficiency.

To quantify the diversity contribution of each individual instance, following (Wang et al., 2026), we first obtain a 2-dimensional representation for each function by extracting the ELA features of each function using the *pflacco* package (Prager & Trautmann, 2023) and applying them to a pre-trained autoencoder provided by (Wang et al., 2026) [1] to derive a hidden representation $\mathbf{z}$. Then, we compute the averaged Euclidean distance from each problem instance to its $k$ nearest neighbors in the standardized representation space as the diversity score. Formally, the diversity score $S_i$ for the $i$-th problem instance is calculated by:

$$D_i = \frac{1}{k} \sum_{j \in \mathcal{N}_i} \|\mathbf{z}_i - \mathbf{z}_j\|_2,$$

where $\mathbf{z}_i$ is the normalized feature vector of the $i$-th instance, and $\mathcal{N}_i$ represents the set of indices of its $k$ nearest neighbors. A higher score $D_i$ indicates that the instance resides in a sparse region of the feature space, thereby contributing more to the overall diversity.

Additionally, to strictly quantify the quality of a problem instance $f$ for MetaBBO, we measure its marginal contribution to the generalization capability of the learned policy $\pi_\phi$. Given a finite training set $\mathcal{D}_{\text{train}} = \{f_1, \ldots, f_N\}$ and a held-out test set $\mathcal{D}_{\text{test}}$, we adopt a Leave-One-Out (LOO)

---

[1] https://github.com/MetaEvo/Diverse-BBO/blob/main/models/autoencoder_epoch_300.pth

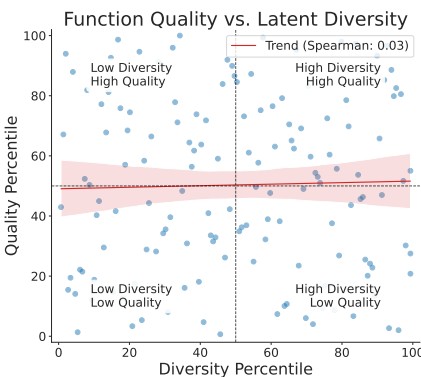

*Figure 2.* Scatter plots showing the correlation between function quality and its latent diversity. Each point corresponds to one function instance, with diversity measured in the latent space of ELA feature and quality quantified by its LOO contribution. The near-zero Spearman correlation (0.03) indicates a weak relationship between function quality and diversity.

valuation protocol. For each instance $f_i \in \mathcal{D}_{\text{train}}$, we train a MetaBBO agent on the subset $\mathcal{D}_{-i} = \mathcal{D}_{\text{train}} \setminus \{f_i\}$ to obtain the marginal optimal meta-parameters $\phi^*_{-i}$. The quality score $Q_i$ is formally defined as the performance gap on $\mathcal{D}_{\text{test}}$ between the policy trained on the full dataset and the policy trained without $f_i$:

$$Q_i = \mathbb{E}_{f \sim \mathcal{D}_{\text{test}}} \left[ \mathcal{M}(\mathcal{A}(\pi_{\phi^*_{\text{all}}}, f)) \right]$$
$$- \mathbb{E}_{f \sim \mathcal{D}_{\text{test}}} \left[ \mathcal{M}(\mathcal{A}(\pi_{\phi^*_{-i}}, f)) \right],$$

where $\phi^*_{\text{all}}$ denotes the optimal meta-parameters learned from the full training set $\mathcal{D}_{\text{train}}$. A higher value of $Q_i$ indicates that instance $f_i$ contains critical features that are essential for the optimizer $\mathcal{A}$ to generalize to unseen problems, thereby signifying high quality.

In practice, we use the function instances set constructed by (Wang et al., 2026) [2], which contains 256 diverse instances and encompasses a wide coverage of the ELA features, to train SYMBOL (Chen et al., 2024). We evaluate the model's performance on the held-out test instance set from (Wang et al., 2026), which consists of unseen synthetic functions and realistic optimization problems including HPO-B (Arango et al., 2021), protein (Tsaban et al., 2022) and UAV (Shehadeh & Kudela, 2025).

In Figure 2, we visualize the scatter plots of the instances' diversity and quality and calculate their Spearman correlation coefficient, where the data points are uniformly distributed across the entire space, occupying all four quadrants. Besides, the extremely low Spearman correlation (0.03) reveals a decoupled nature between these two metrics. This observation underscores the necessity of explicitly incorporating

---

quality-aware guidance into the training instances generation, rather than relying solely on diversity maximization, motivating us to propose a novel quality-oriented data generation framework for MetaBBO.

### 3.2. Performance Gap-based Utility of Function Instances for MetaBBO

The empirical analysis in Section 3.1 establishes that high feature diversity does not guarantee high training quality ($Q_i$). Ideally, an effective data generator should prioritize instances with high $Q_i$. However, computing the marginal contribution $Q_i$ via the LOO protocol is computationally prohibitive during the training process, as it requires retraining the model from scratch for every candidate instance. To make quality-aware generation feasible in online MetaBBO agent training, we replace this expensive global LOO objective with a simple proxy: maximizing the instantaneous utility of each instance with respect to the current agent, following common practices in RL curriculum design (Portelas et al., 2020; Bengio et al., 2009; Schaul et al., 2015).

In general RL, the utility of a training environment is typically characterized by *regret* (Jiang et al., 2021b; Dennis et al., 2020), which measures the shortfall of the agent's policy against a theoretical optimal policy. However, in the context of BBO, relying on the theoretical global optimum is flawed. Effective optimization requires underlying landscape structure (Jones & Forrest, 1995; Mersmann et al., 2011); yet, randomly generated functions often exhibit chaotic or deceptive landscapes, effectively acting as unoptimizable noise where the global optimum is practically indistinguishable from random chance (Wolpert & Macready, 1997). In such intractable scenarios, a large global regret reflects the inherent difficulty of the problem rather than the agent's incompetence. Consequently, defining utility based on global optimum creates a misalignment that encourages the agent to chase unsolvable noise.

To mitigate the effects of problem difficulty, we instead assess the agent's *relative potential for improvement* against a competent baseline (e.g., CMA-ES (Hansen et al., 2003) or MadDE (Biswas et al., 2021)). Intuitively, an instance is most informative when the agent underperforms compared to the teacher baseline, indicating that the instance is solvable but not yet mastered by the current MetaBBO policy $\pi_\phi$. Formally, we define the utility as the *Regret Gap* for MetaBBO by:

$$U(f, \phi) = \log \mathcal{M}(\mathcal{A}_{\text{baseline}}(f)) - \log \mathcal{M}(\mathcal{A}(\pi_\phi, f)). \quad (1)$$

This gap serves as a precise learning signal. A near-zero or negative $U$ suggests the task is already learned (low utility), whereas an excessively large $U$ implies the task might be too difficult with respect to the agent. In practice, we employ the concept of "zone of proximal development" (Vygotsky,

1978; Wang et al., 2022) to prioritize instances within a constructable range $[u_{\min}, u_{\max}]$. This reformulation casts data generation as a *dynamic curriculum learning* problem: Rather than training on a static or random sequence of functions, we construct a curriculum that balances the exploration of high-utility new tasks with the replay of critical historical instances to prevent forgetting.

To facilitate this learning process, a well-constructed structure for BBO functions is required, since navigating the unstructured space of all possible mathematical functions is inefficient. In the next subsection, we will introduce a structured parameterization that allows us to effectively select and schedule candidate instances.

### 3.3. Instance Structures of BBO Functions

To enable a structured curriculum, we first revisit the core principles governing BBO problem difficulty, which then motivate us to construct hierarchical generative space.

**Principles of Landscape Construction.** In BBO, the difficulty of a problem instance is not random but stems from specific landscape characteristics (Hansen et al., 2021; Mersmann et al., 2011). These characteristics can be broadly categorized into two distinct sources:

- *Global Topology*: This refers to the fundamental topological shape of the objective function, e.g., convexity, the presence of local optima, or the existence of a global funnel structure (Mersmann et al., 2011). These properties define the type of search strategy required (e.g., exploration or exploitation) (Lunacek et al., 2008), indicating inherent hardness.

- *Geometric Distortion*: Given a fixed topology, the difficulty can vary dynamically based on coordinate transformations. Factors such as high condition numbers, rotation, and shifting distort the landscape, varying external difficulty testing BBO optimizers' invariance and adaptation capabilities (Hansen et al., 2021; Song et al., 2022).

A robust learning curriculum must systematically cover both diverse topologies and varying degrees of distortion. Based on these principles, we formalize the BBO instance space as a hierarchical generative space.

**Hierarchical Generative Space.** We instantiate the generation of BBO instances as a two-stage hierarchical procedure. An illustrative example of generating a bowl-shaped instance is shown in Figure 3, where we first determine functions' global topology by setting its base class, and then apply detailed distortion (e.g., affine transformation) to obtain a concrete instance. Formally, denoted as $\mathcal{G} = \langle \mathcal{C}, \Theta \rangle$, the formulated hierarchical generative space consists of two stages of instance generation:

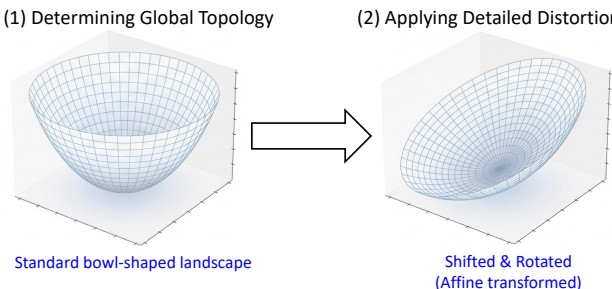

(1) Determining Global Topology     (2) Applying Detailed Distortion

Standard bowl-shaped landscape     Shifted & Rotated (Affine transformed)

*Figure 3.* Illustration of the hierarchical generation of a bowl-shaped instance.

- Stage 1: Topological class space $\mathcal{C}$ generation. Let $\mathcal{C} = \{c_1, \ldots, c_K\}$ be a discrete set of base function classes (e.g., Sphere, Rastrigin, Ackley). Selection at this level determines the global topology. By switching between classes in $\mathcal{C}$, the curriculum exposes the model to fundamentally different optimization landscapes, preventing overfitting to a single problem type (Chen et al., 2024). Notably, simply mixing function class (e.g., via affine transformation) could lead to new topologies (Vermetten et al., 2025).

- Stage 2: Distortion parameter space $\Theta$ generation. Given a class $c$, the continuous parameter space $\Theta$ defines the transformations applied to the base function. Taking affine transformation as an example, a parameter vector $\boldsymbol{\theta} \in \Theta$ typically includes rotation matrices $\mathbf{R}$, scaling factors $\boldsymbol{\Lambda}$, and shift vectors $\mathbf{s}$. Selection at this level determines the *local distortion*, allowing the curriculum to adjust the difficulty within a specific topology (Hansen et al., 2021).

Then, a specific training instance is generated by sampling a class $c$ and parameters $\boldsymbol{\theta}$, then constructing:

$$f(\mathbf{x}; c; \boldsymbol{\theta}) = g_c\big(\mathcal{T}(\mathbf{x}; \boldsymbol{\theta})\big),$$

where $g_c$ represents the canonical base function and $\mathcal{T}$ applies the geometric transformations. This hierarchical structure naturally motivates the design of our framework, which will be introduced in the next section.

## 4. An Efficient Data Generation Framework via Hierarchical Active Task Selection

In this section, we present our framework, Hierarchical Active Task Selection (HATS), which incorporates a bi-level strategy motivated based on the hierarchical generative space discussed in Section 3.3. As shown in Figure 4, HATS comprises: 1) a bandit-based mechanism to allocate attention across different function classes $\mathcal{C}$, and 2) an active selection mechanism combining exploration and replay exploitation to schedule the parameters $\boldsymbol{\theta}$ efficiently. One

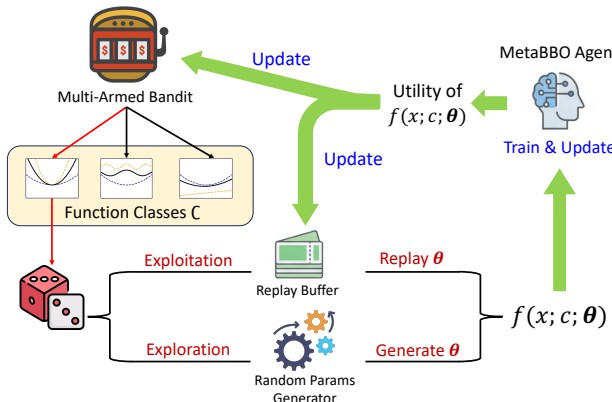

*Figure 4.* Illustration of our proposed HATS framework.

key characteristic of HATS is that the generator is updated *online* using the agent's current competence, rather than optimizing a static diversity objective.

### 4.1. Multi-Armed Bandit for Function Class Selection

HATS first employs a Multi-Armed Bandit (MAB) mechanism to dynamically allocate the training budget across $K$ distinct function classes, where we treat each function class $c_k \in \mathcal{C}$ (where $K = |\mathcal{C}|$) as an arm. We maintain a logit vector $\boldsymbol{\ell}^{(t)} = [\ell_1^{(t)}, \ldots, \ell_K^{(t)}]$ to track the time-varying utility of each class. Unlike standard stationary bandits, the utility here depends on the agent's evolving competence, showing more robust response to the high stochasticity of the feedback signal.

At each training step $t$, we determine the selection probability $p_k^{(t)}$ for each class $c_k$. To prevent starvation of temporarily difficult classes and ensure continuous exploration, we employ a mixture policy that combines a Softmax distribution with uniform exploration. The probability is defined as:

$$p_k^{(t)} = (1 - K\epsilon) \cdot \frac{\exp(\ell_k^{(t)})}{\sum_{j=1}^{K} \exp(\ell_j^{(t)})} + \epsilon,$$

where $\epsilon = 0.05$ is a small constant ensuring a non-zero floor probability for every class. Based on $\boldsymbol{p}^{(t)}$, a batch of task categories is sampled and subsequently populated with concrete instances by the instance generator.

However, empirically, we find that deciding the function class solely based on simple MAB would lead to probability collapse, as shown in Figure 5. During the training of SYM-BOL (Chen et al., 2024) on 10 CEC function classes (Mohamed et al., 2021), we find that the agent struggles in the bi_Rastrigin function. As training progresses, the probability of selecting bi_Rastrigin converges to 0.55, the theoretical maximum probability of a function class, i.e., the MAB collapses to this arm. This results in poor performance for the agent. We can observe from the right sub-figure of

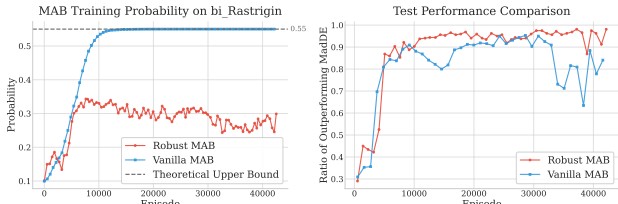

*Figure 5.* Learning dynamics of Robust MAB and Vanilla MAB. The left sub-figure plots MAB probability of choosing the bi_Rastrigin function, while the right one displays the test performance through training.

Figure 5 that after 30000 episodes, the agent trained with vanilla MAB suffers from a catastrophic forgetting. To mitigate this, we employ two techniques to prevent MAB's collapsing, which we termed Robust MAB, including:

- **Rank-Based Utility Feedback.** After the agent trains on the sampled batch, we quantify the value of each class via *Rank Utility*. For a batch of $N$ tasks, we calculate the utility of each instance $i$ using Equation (1), rank them within the batch, and assign a normalized score $v_i \in [0, 1]$ based on the ranking. The feedback $\bar{v}_k^{(t)}$ for class $c_k$ is the mean utility of all instances belonging to that class in the current batch.

- **Robust Update Mechanism.** To enable a stable update of the logits, we employ a centered decaying update rule, which is given by:

$$\ell_k^{(t+1)} = \text{clip}\left(\gamma \cdot \ell_k^{(t)} + \eta \cdot (\bar{v}_k^{(t)} - 0.5), \ -L, \ L\right).$$

Here, we center the feedback to 0 by subtracting 0.5, ensuring that classes performing below the median are penalized while those above are rewarded. The decay factor $\gamma \in (0, 1)$ induces a forgetting mechanism for outdated priors, allowing the scheduler to adapt to the agent's new capabilities.

In Figure 5, we can observe that compared to Vanilla MAB, Robust MAB assigns a more balanced probability to the bi_Rastrigin arm, showing more smooth and consistent improvement on the test set throughout training.

### 4.2. Active Parameter Sampling via Utility-Aware Replay

Once the MAB allocates the training budget to a specific function class $c$, we employ a Utility-Aware Replay (UAR) strategy to determine the specific parameters $\boldsymbol{\theta}$. While building upon the Prioritized Level Replay (PLR) framework (Jiang et al., 2021b), our mechanism is tailored to select instances from a global buffer $\mathcal{B}$ that maximize training efficiency for the chosen class. The overall procedure is outlined in Algorithm 1.

**Algorithm 1** UAR for Instance Parameters

---

1: **Input:** Function class $c$, global buffer $\mathcal{B}$, policy $\pi_\phi$
2: Sample replay decision $d \sim$ Bernoulli($\rho$)
3: **if** $d = 0$ **or** no valid tasks for $c$ in $\mathcal{B}$ **then**
4:     // Exploration: Generate new task, no gradient update
5:     Sample parameters $\theta$ from generator for class $c$
6:     Collect trajectory $\tau$ on $\theta$ with *stop-gradient* $\phi_\perp$
7: **else**
8:     // Replay: Sample based on Score $S$ (Equation (2))
9:     Sample $\theta \sim \mathcal{B}$ restricted to class $c$ using prob. $\boldsymbol{p}$
10:    Collect trajectory $\tau$ on $\theta$ and update $\phi$ with rewards $R(\tau)$
11: **end if**
12: Compute utility $u = U(\tau, \pi)$ via Equation (1)
13: Update global buffer $\mathcal{B}$ with instance $\theta$ and utility $u$

---

**Prioritized Sampling Strategy.** To select the most effective training tasks from the global buffer $\mathcal{B}$, we calculate a sampling score $S_i$ for each instance $i$ belonging to class $c$. Let $u_i$ denote the utility score derived from Equation (1) and $t_i$ denote the staleness (number of steps since the last replay). The score is computed as:

$$S_i = \frac{u_i - u_{\min}}{u_{\max} - u_{\min}} + \alpha \cdot \frac{t_i - t_{\min}}{t_{\max} - t_{\min}},$$

where values are normalized over the valid subset of the buffer. The hyperparameter $\alpha$ balances the trade-off between utility and staleness. The probability of replaying instance $i$ is then determined by a Softmax distribution with temperature $\tau$:

$$p_i = \frac{\exp(S_i/\tau)}{\sum_{j \in \mathcal{B}_c} \exp(S_j/\tau)}, \qquad (2)$$

where $\mathcal{B}_c$ denotes the subset of the global buffer containing instances of class $c$, and $\tau$ is set to 1.0. This ensures that the agent focuses on tasks with high training utility while periodically revisiting older instances to prevent catastrophic forgetting (Jiang et al., 2021b).

Additionally, as recommended by (Jiang et al., 2021a), UAR employs a *stop-gradient* operation $\phi_\perp$ to exploration trajectories, updating the policy $\pi$ only on verified replay instances. As detailed in Algorithm 1, when exploring new parameters (Line 4), we generate $\theta$ from the function class $c$ but strictly suppress the policy update. Gradient updates are performed only during the replay phase (Line 8), ensuring the agent learns only from tasks with high utility.

# 5. Experiments

In this section, we empirically study our proposed method, HATS, from different perspectives. We introduce the experimental setup and the compared tasks in Section 5.1. Then

we show the performance of HATS and answer several Research Questions (RQs) in Section 5.2.

## 5.1. Experimental Setup and Selected Tasks

**MetaBBO Backbone.** We build upon SYMBOL (Chen et al., 2024), a population-based MetaBBO agent that integrates symbolic equation learning to learn the update rule for BBO. We provide a detailed introduction of the SYMBOL backbone in Appendix A.1. Notably, in the original implementation, SYMBOL uses CEC functions (Mohamed et al., 2021) with DR augmentation to train the agent. For implementation, we modify the data generation part and keep other parts unchanged.

**Compared Baselines.** We compare HATS against two categories of baselines: (1) **Static training**: Models trained on fixed function sets without augmentation or curriculum mechanisms. This includes two standard benchmarks, BBOB (Hansen et al., 2021), which has 24 distinct base functions, and CEC (Mohamed et al., 2021), which consists of 10 function classes. We use recent function generation methods, MA-BBOB (Vermetten et al., 2025) and Diverse-BBO (Wang et al., 2026); (2) **DR** (Tobin et al., 2017): A widely adopted baseline that applies random shifts and rotations to augment the base functions (i.e., functions from BBOB and CEC), obtaining diverse instances. Details of baselines implementation, as well as hyperparameter settings of HATS, can be found in Appendix B.

**Tasks.** The benchmark tasks used for comparing all methods comprises two parts. The first part covers unseen synthetic instances constructed via base functions from CEC (Mohamed et al., 2021) and BBOB (Hansen et al., 2021) with unseen shifts and rotation. We also consider realistic optimization tasks from MetaBox (Ma et al., 2023; 2025c) for evaluation, including Protein Docking (Tsaban et al., 2022) and UAV (Shehadeh & Kudela, 2025) [3]. These selected tasks cover a wide range of 1,424 distinct instances to assess all methods' performance.

## 5.2. Experiment Results

In this section, we present our experimental findings, aiming to answer the following RQs.

**RQ1: Does HATS improve generalization of the learned MetaBBO agent?** In Table 1, we report the performance of HATS compared to the baselines on the selected tasks, where we can observe that: (1) HATS achieves state-of-the-art performance by obtaining an average rank of 1.25 and 2.25 when trained on BBOB and CEC functions, respectively. Notably, HATS consistently outperforms DR except

---

[3]We exclude HPO-B (Arango et al., 2021) tasks from MetaBox due to HPO-B's unstable surrogate evaluation, as demonstrated by (Müller et al., 2023).

*Table 1.* Normalized performance on CEC, BBOB, Protein Docking and UAV, where the best and runner-up on each task are **Blue** and **Violet**. The results are scaled to the range [0, 1], and lower values indicate better performance.

| Base Function | Scheduler Type | CEC | BBOB | Protein Docking | UAV | Avg. Score | Avg. Rank |
|---|---|---|---|---|---|---|---|
| MA-BBOB | Static | 0.249 ± 0.204 | 0.236 ± 0.189 | 0.703 ± 0.099 | 0.427 ± 0.261 | 0.404 | 6.50 |
| Diverse-BBO | Static | 0.312 ± 0.195 | 0.374 ± 0.156 | 0.330 ± 0.069 | 0.478 ± 0.259 | 0.373 | 7.00 |
| BBOB | Static | 0.260 ± 0.172 | 0.268 ± 0.131 | 0.440 ± 0.107 | 0.483 ± 0.278 | 0.363 | 7.00 |
| | DR | 0.122 ± 0.106 | 0.145 ± 0.112 | 0.189 ± 0.061 | 0.331 ± 0.219 | 0.197 | 3.75 |
| | **HATS (Ours)** | **0.041 ± 0.053** | **0.068 ± 0.129** | 0.150 ± 0.069 | **0.066 ± 0.156** | **0.081** | **2.25** |
| CEC | Static | 0.181 ± 0.127 | 0.173 ± 0.109 | 0.569 ± 0.114 | 0.279 ± 0.220 | 0.300 | 5.25 |
| | DR | 0.122 ± 0.102 | 0.158 ± 0.131 | **0.112 ± 0.045** | 0.259 ± 0.248 | 0.163 | 3.00 |
| | **HATS (Ours)** | **0.034 ± 0.040** | **0.051 ± 0.048** | **0.116 ± 0.056** | **0.057 ± 0.151** | **0.065** | **1.25** |

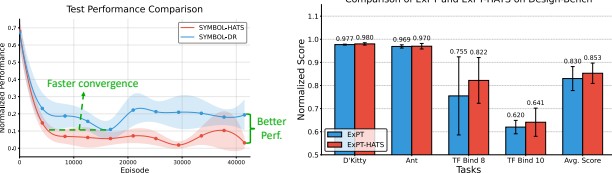

*Figure 6.* Left: Normalized performance (minimization) comparison between SYMBOL-HATS (red) and SYMBOL-DR (blue). Right: Normalized score (maximization) on Design-Bench (Trabucco et al., 2022) showing HATS's improvement on ExPT.

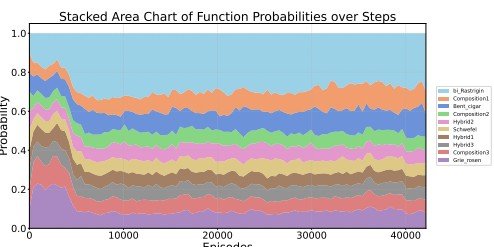

*Figure 7.* Stacked area chart of function class probability distribution of CEC functions during training.

on case, showing the superiority of HATS to improve the generalization ability for the MetaBBO agent. (2) DR performs better than the static baselines, showing that simply augmenting instance diversity could bring improvements than static training.

**RQ2: Does HATS improve training efficiency?** Next, we verify whether HATS improves the training efficiency by plotting test performance against training steps. We plot the normalized test performance on the CEC functions during training in the left sub-figure of Figure 6, where SYMBOL combined with HATS demonstrates faster convergence and better final performance than that with DR. We also provide a case study of these two agents in Appendix C.2.

**RQ3: How does the curriculum evolve under HATS?** A key advantage of HATS is that its curriculum is not manually designed; instead it emerges from utility-based feedback. To better understand the mechanism behind HATS, we investigate the evolution of sampling probabilities for different CEC function classes throughout training in Figure 7. We find that as the training progresses, the probability of some function classes, especially `Grie_rosen` which has a simple valley-like topology, increases in the initial stage and decreases after 5000 episodes, indicating that the agent has learned this landscape. In the subsequent period, the distribution shifts substantially to `bi_Rastrigin`, a hard multi-modal function class. These results clearly show that HATS assigns a dynamic curriculum for MetaBBO training.

**RQ4: Can HATS generalize to other learning-based BBO scenarios?** We additionally examine the effectiveness

of HATS on the few-shot data-driven offline BBO scenario. We build upon ExPT (Nguyen et al., 2023), a framework that directly models the final solution conditioned on the few-shot data. ExPT is pretrained under Gaussian processes with varying hyperparameters. We apply HATS to schedule these hyperparameters. Details of ExPT and our HATS implementation can be found in Appendix B.3. Results in the right sub-figure of Figure 6 show that combined with HATS, the performance of ExPT can be further enhanced, showing the versatility of HATS.

**Additional analysis.** We provide additional results in Appendix C. We first ablate key components of HATS, i.e., MAB for task class allocation (Section 4.1) and UAR for parameter scheduling (Section 4.2) in Appendix C.1, which validates the effectiveness of each component. In Appendix C.2 and C.3, we provide a case study of comparing HATS and DR, and display the optimization curve of each type of tasks, respectively.

## 6. Conclusion

In this work, we focus on the data generation issue for MetaBBO. We identify the Diversity-Quality Gap in MetaBBO and propose HATS, a framework that bridges this gap by dynamically selecting curriculum based on a regret-based utility metric. Empirical studies show the superiority and versatility of HATS. Future work may extend HATS to a generator that not only schedules instances but also creates effective data iteratively, e.g., via generating symbolic functions (Meidani et al., 2024; Wang et al., 2025).

## Impact Statement

This paper presents HATS, whose goal is to advance the field of meta-black-box optimization by introducing a principled data generation framework. There are many potential societal consequences of our work, none which we feel must be specifically highlighted here.

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

## A. A Brief Introduction of the Employed Method Backbone

### A.1. SYMBOL

SYMBOL (Chen et al., 2024) is a MetaBBO framework that learns optimizers by symbolically generating update rules. Instead of using a fixed hand-crafted algorithm, SYMBOL trains a symbolic equation generator, parameterized by RL policy, to output an explicit update expression built from a predefined set of basic operators and optimization-related operands. The generated symbolic expression is then executed as the optimizer's update rule to propose new candidate solutions during search, which yields an adaptive yet interpretable optimization procedure.

Practically, the equation generator is meta-trained over the CEC function class (Mohamed et al., 2021), which is augmented with random shift and rotation. The policy is trained via Proximal Policy Optimization (Schulman et al., 2017). The reward consists of two parts: A base reward indicating the optimizer's absolute performance and a guided reward forcing the optimize the mimic the behavior of a teacher algorithm, typically instantiated by MadDE (Biswas et al., 2021).

### A.2. ExPT

ExPT (Nguyen et al., 2023) is a pretraining-adaptation framework for few-shot offline BBO, i.e., optimizing the black-box function $f$ using only a few-shot dataset $\{\mathbf{x}_i, f(\mathbf{x}_i)\}_{i=1}^N$. ExPT pretrains a transformer model (Vaswani et al., 2017) with a large amount of synthetic data, so that it can adapt to a new objective using only a few labeled observed $(x, y)$ pairs. ExPT uses *inverse modeling*: given a small set of context points $(\mathbf{x}_{1:m}, y_{1:m})$ and a set of desired target outputs $y_{m+1:N}$, the model approximates $p(\mathbf{x}_{m+1:N} \mid \mathbf{x}_{1:m}, y_{1:m}, y_{m+1:N})$, and samples candidate $\mathbf{x}$ during inference.

To obtain substantial pretraining data, ExPT samples synthetic functions from a Gaussian process prior with an RBF kernel (Rasmussen & Williams, 2006), $\tilde{f} \sim \mathcal{GP}(0, K)$ where $K(x, x') = \sigma^2 \exp\left(-\frac{\|x-x'\|^2}{2\ell^2}\right)$. Here $\sigma$ (variance) controls the scale of function values and $\ell$ (length scale) controls smoothness. Specifically, ExPT randomizes $\sigma$ and $\ell$ to increase function diversity. For each sampled function, it evaluates $\tilde{f}$ on a set of unlabeled inputs, then splits the resulting input-output pairs into a small context set and a target set to form training examples.

## B. Implementation Details

### B.1. Detailed Setting and Hyperparameters for HATS

To quantify the utility performance gap in Equation (1), following SYMBOL to select MadDE (Biswas et al., 2021) as a learnable teacher, we use MadDE as the baseline algorithm to evaluate the hardness of the generated instances. Other hyperparameters for HATS are listed in Table 2.

*Table 2.* Hyperparameter setting for HATS.

| Parameter Name | Values |
|---|---|
| Utility range $[u_{\min}, u_{\max}]$ | [-0.5, 4.0] |
| Decay rate $\gamma$ for bandit logits | 0.9 |
| Update rate $\eta$ for bandit logits | 0.1 |
| Clipping bound $L$ for bandit logits | 1.5 |
| Balancing coefficient $\alpha$ for staleness in replay buffer | 0.1 |
| Optimizer | Adam (Kingma & Ba, 2015) |
| Learning rate | $1 \times 10^{-3}$ |

### B.2. Implementation Details of Compared Baselines

For BBOB functions, we use the inference from MetaBox (Ma et al., 2025c) (https://github.com/MetaEvo/MetaBox/tree/v2.0.0/src/environment/problem/SOO/COCO_BBOB).

For CEC functions, we directly use the implementation provided by SYMBOL (Chen et al., 2024) (https://github.com/MetaEvo/Symbol/blob/main/dataset/cec_dataset.py).

For MA-BBOB (Vermetten et al., 2025), we use the implementation by IOHProfiler (Doerr et al., 2019) by randomly selecting a set of 24 function instances and train the agent using this set.

For Diverse-BBO, we directly use the `pickle` file provided by (Wang et al., 2026) (`https://github.com/MetaEvo/Diverse-BBO/blob/main/save_gp_functions/all_256_programs.pickle`). However, we find that the provided function instances are prone to outliers due to the symbolic function property, which may cause damage to the training process. Therefore, we take a dynamic strategy that once we meet an outlier that cause numerical overflow, we filter out the current instance and resume the checkpoint from last episode to enable training for Diverse-BBO.

## B.3. Implementation Details of HATS on ExPT

At each pretraining iteration, we sample GP hyperparameters $\theta$ to generate synthetic functions and train ExPT on the collected data from the functions. We define the utility of a sampled configuration $\theta$ by the training negative log-likelihood: $U_{\text{ExPT}}(\theta) = -\log p_\phi(\mathbf{x}_{m+1:N} \mid \mathbf{x}_{1:m}, y_{1:m}, y_{m+1:N})$ where $p_\phi$ denotes the distribution parameterized by the model, and use it as the feedback signal for utility-aware replay. This follows the HATS principle that $U$ should reflect the training value of an instance for the current model.

In our implementation, we apply HATS by scheduling the two core GP hyperparameters that govern instance generation and difficulty: the lengthscale $\ell$ and the function scale $\sigma$. Let $\theta = (\ell, \sigma)$, $\ell \sim \mathcal{U}[1.0, 10.0]$, $\sigma \sim \mathcal{U}[5.0, 10.0]$. These prior ranges are set following the ExPT. We perform utility-aware replay over previously sampled $\theta$ for training.

Following ExPT, we evaluate the trained model on 4 tasks from Design-Bench (Trabucco et al., 2022), a prevalent benchmark suite for offline BBO. These tasks include 2 continuous tasks, Ant Morphology (Brockman et al., 2016) and D'Kitty Morphology (Ahn et al., 2020), and 2 discrete tasks, TF-Bind-8 and TF-Bind-10 (Barrera et al., 2016). The objective of Design-Bench is to maximize the objective score, which is different to our main experiment in Table 1.

# C. Additional Experimental Results

## C.1. Ablation Study: Does Each Component of HATS Contribute to the Final Performance?

We ablate key components in HATS to isolate their contributions by: (i) removing the multi-armed bandit for function class selection, (ii) removing utility-aware replay for instance parameters. Experimental results in Figure 8 clearly demonstrates the effectiveness of each component in HATS.

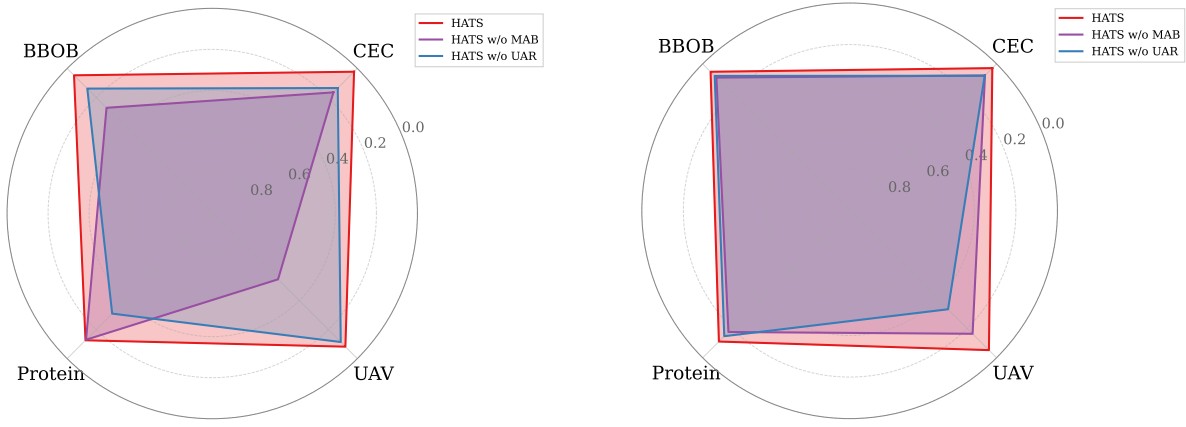

*Figure 8.* Ablation results on HATS components, MAB and UAR.

## C.2. Case Study: How does SYMBOL Combined with HATS Generalize Better than DR?

Figure 9 compares the behaviors of SYMBOL-DR and SYMBOL-HATS on an unseen shifted-and-rotated 2D Rastrigin instance. In the left panel, SYMBOL-HATS reaches the true optimum within fewer iterations, and its trajectory is more clustered with fewer large cross-region jumps, indicating a more stable search behavior. In contrast, in the right panel,

SYMBOL-DR repeatedly searches around $(0, 0)$ and makes frequent long-range jumps across the domain, failing to reach the true optimum. Consistent with these observations, the bottom curve shows that SYMBOL-HATS reduces the objective faster, whereas SYMBOL-DR decreases more slowly and levels off at a worse objective.

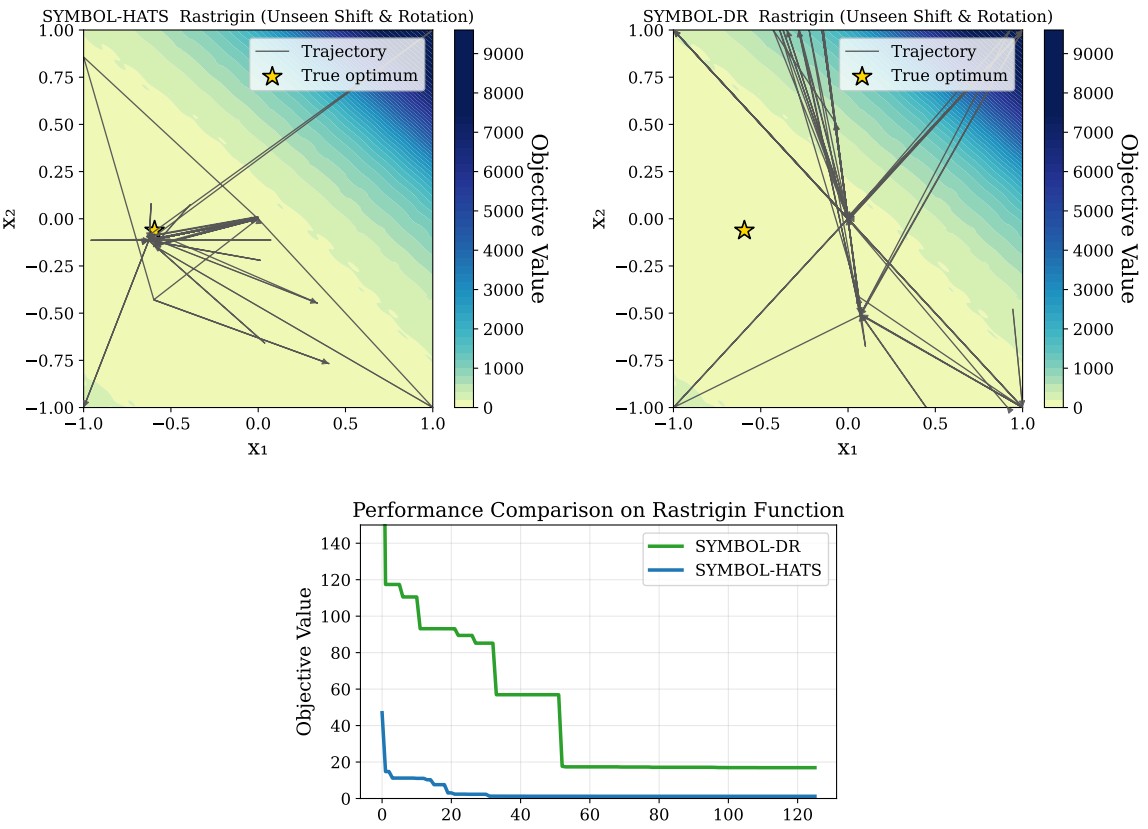

*Figure 9.* Performance Comparison on Rastrigin: SYMBOL trained with HATS (termed as SYMBOL-HATS) versus SYMBOL-DR. The top panels visualize the 2D objective landscape of an unseen shifted-and-rotated Rastrigin function with optimization trajectories produced by SYMBOL-HATS (left) and SYMBOL-DR (right), and the bottom panel compares their objective values across generations. SYMBOL-HATS exhibits more stable search behavior.

### C.3. Optimization Curve

In Figures 10 to 12, we visualize the optimization curve of all compared methods on BBOB, CEC, Protein and UAV. Note that for BBOB and CEC, we augment the testing instances by randomly sampling unseen parameters for testing, we plot the detailed performance of each function by averaging objective value at each iteration. For protein docking and UAV, we visualize the average performance over all test problems. The objective value are normalized to [0,1] with respect to each instance. These results clearly show the superiority of our proposed method, HATS.

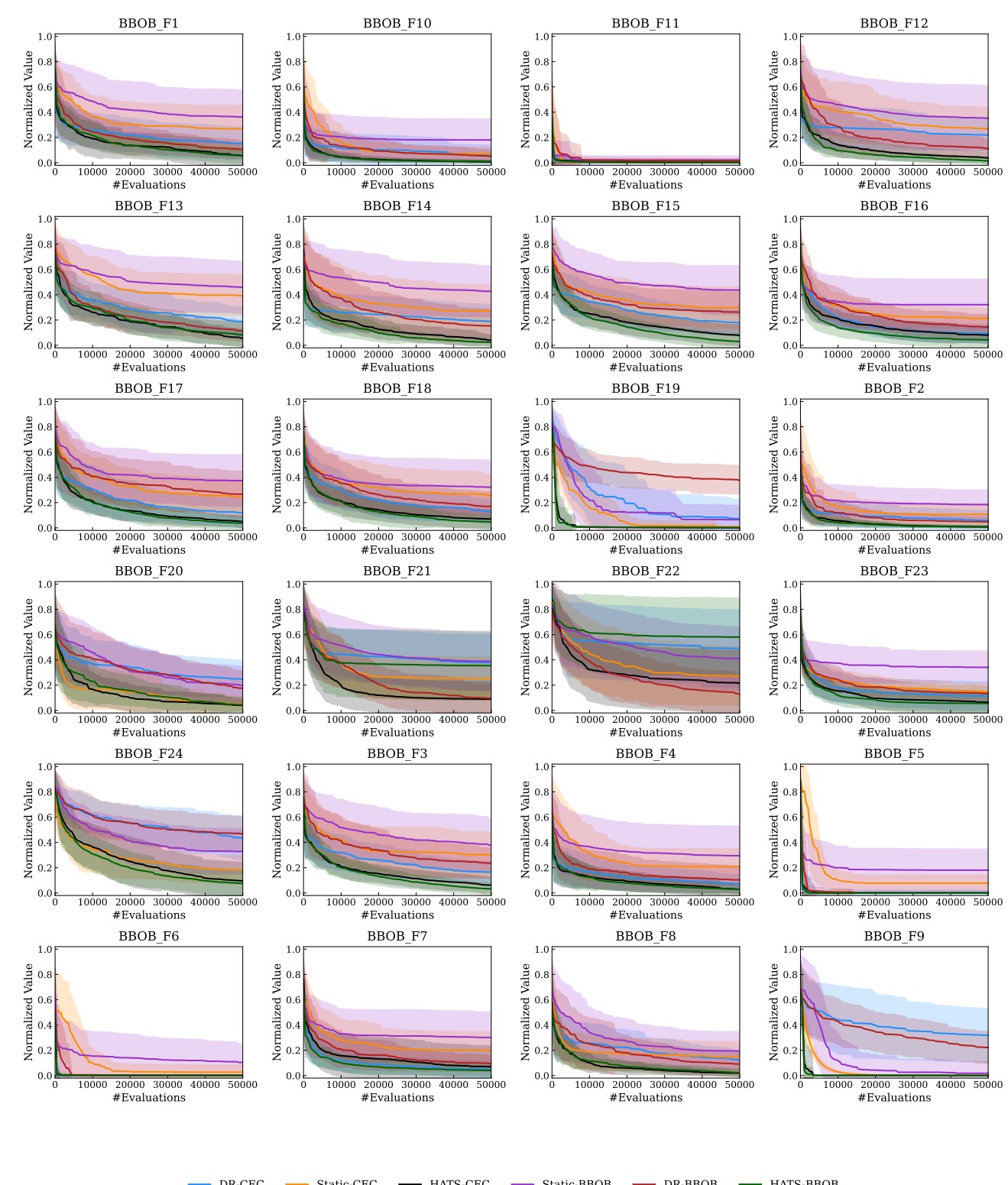

*Figure 10.* Optimization curve of different methods on BBOB.

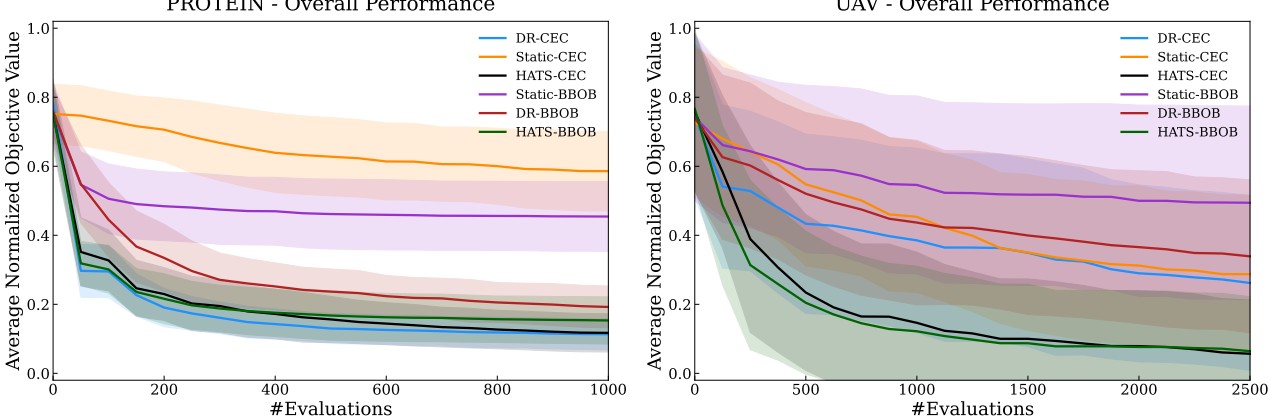

*Figure 11.* Optimization curve of different methods on CEC.

*Figure 12.* Optimization curve of different methods on Protein Docking and UAV.

