# OpenReview forum: "Principled Data Generation for MetaBBO via Active Task Selection"
_ICML.cc/2026/Conference — Submitted to ICML 2026_

### Official Review · Reviewer_duoh · 2026-03-06

**Soundness:** 2
**Presentation:** 2
**Significance:** 3
**Originality:** 3
**Overall Recommendation:** 4
**Confidence:** 4

**Summary:**

This paper introduces Hierarchical Active Task Selection (HATS), a training task generation framework designed for Meta-Black-Box Optimization (MetaBBO). HATS employs a two-stage generation process: in the first stage, a multi-armed bandit (MAB) mechanism selects a function class based on time-varying utilities.  In the second stage, an utility-aware
replay (UAR) strategy determines the distortion parameters (e.g. scaling factors and rotation matrices) for the chosen class by balancing exploration and exploitation, thereby synthesizing specific training instances. Empirical results show that HATS increases the training efficiency of MetaBBO and the performance of the low-level optimizers including SYMBOL and ExPT.

**Compliance With Llm Reviewing Policy:**

Affirmed.

**Key Questions For Authors:**

(1) How do MetaBBO backbones equipped with HATS perform compared to classic BBO methods like CMA-ES? (2) Does the authors investigate the performance improvement of a wider range of MetaBBO backbones equipped with HATS? (3) Could the authors provide more details about the dataset (e.g. the size of training sets, dimensional settings for training and testing sets)?

**Limitations:**

Please refer to Weakness (1) and (4).

**Strengths And Weaknesses:**

**Strengths** (1) This paper propose an effective training task generator for MetaBBO,   experiment results show that HATS improve the relative peformance of low-level learnable black-box optimizers. (2) HATS reveals that training efficiency is not strongly correlated with the diversity of training tasks, but is related to their quality (e.g., the utility for improving the low-level optimizer's performance).

**Weaknesses** (1) The experimental section only involves the metabbo methods equipped with HATS and their vanilla versions, but it does not include a comparison with classic BBO methods (e.g. CMA-ES) to show more intuitive performance changes. (2) None of the equations in the paper are numbered, which makes reading somewhat difficult. (3) The details of the training sets are not elaborated (e.g. the size and dimensional settings of the training sets). (4) Including more MetaBBO backbones can make the effectiveness of HATS more convincing.

 e.g.

Lange, Robert, et al. "Discovering evolution strategies via meta-black-box optimization." Proceedings of the companion conference on genetic and evolutionary computation. 2023.

Li, X., Wu, K., Zhang, X., & Wang, H. (2025, April). B2opt: Learning to optimize black-box optimization with little budget. In Proceedings of the AAAI Conference on Artificial Intelligence (Vol. 39, No. 17, pp. 18502-18510).

Chen, J., Ma, Z., Guo, H., Ma, Y., Zhang, J., & Gong, Y. J. (2024). SYMBOL: Generating flexible black-box optimizers through symbolic equation learning. arXiv preprint arXiv:2402.02355.

Li, X., Wu, K., Zhang, X., Wang, H., & Liu, J. (2024). Pretrained optimization model for zero-shot black box optimization. Advances in Neural Information Processing Systems, 37, 14283-14324.

Han, M., Li, X., Wu, K., Zhang, X., & Wang, H. (2025). Enhancing zero-shot black-box optimization via pretrained models with efficient population modeling, interaction, and stable gradient approximation. In The Thirty-ninth Annual Conference on Neural Information Processing Systems.

---

> ### Author Rebuttal · Authors · 2026-03-31
>
> Thanks for your encouraging comments. Below please find our responses. Corresponding experimental results can be found at https://anonymous.4open.science/api/repo/HATS-CA36/file/duoh.pdf.
>
> ## W1\&Q1\&W3\&Q4 Comparison to classical BBO methods and more MetaBBO backbones
>
> Thanks for this helpful comment. We agree that it is important to evaluate HATS on a broader set of MetaBBO backbones and to compare the resulting methods with classical BBO solvers for a more intuitive performance reference.
>
> Following suggestions of both yours and Review QL7R’s, we extend the evaluation to *additional MetaBBO backbones*, including B2Opt [1], POM [2], and LDE [3], beyond SYMBOL. We exclude LES [4] and EPOM [5] since there is no open-sourced implementation. As shown in Fig. R1, the overall trend is consistent across backbones: **HATS > DR > Static** on both synthetic and real-world tasks. This suggests that the benefit of HATS is not tied to a specific training pipeline, but comes more generally from improving the training distribution via active task selection.
>
> We also include *classical BBO baselines*, such as CMA-ES, PSO, and MadDE, for reference. The results show a nuanced picture: on synthetic tasks, classical handcrafted optimizers are competitive, while on real-world tasks, HATS-equipped MetaBBO methods become much more competitive, and POM obtains the best overall results in real-world tasks. Notably, although HATS uses MadDE as the teacher during training, the final learned optimizer can exceed its performance, showing benefits of MetaBBO methods and HATS curriculum.
>
> Overall, these additional results provide stronger evidence that HATS is a largely backbone-agnostic training framework for MetaBBO. We will update these results in the revised version. Thank you very much!
>
> [1] B2opt: Learning to optimize black-box optimization with little budget. AAAI 2025.
>
> [2] Pretrained optimization model for zero-shot black-box optimization. NeurIPS 2024.
>
> [3] Learning adaptive differential evolution algorithm from optimization experiences by policy gradient. IEEE TEvC 2021.
>
> [4] Discovering evolution strategies via meta-black-box optimization. GECCO 2023.
>
> [5] Enhancing zero-shot black-box optimization via pretrained models with efficient population modeling, interaction, and stable gradient approximation. NeurIPS 2025.
>
> ## W2 Un-numbered equations
>
> We will fix them in revision for better readability. Thank you.
>
> ## W3&Q3 Not elaborated details of the training sets
>
> Thank you for pointing this out. Our training set is not a fixed finite dataset, but is sampled *actively* from synthetic benchmark suites. The main synthetic training settings in our experiments cover both *CEC2021-style functions and BBOB*. Specifically, the CEC2021-style suite contains 10 templates, while BBOB contains 24 function classes. The default dimension for synthetic training is 10.
>
> For evaluation on CEC / BBOB, we use held-out instances with fixed seeds for reproducibility. The standard synthetic test setting also uses dimension d=10. For real-world tasks such as Protein and UAV, we use MetaBox [1] split `difficulty=all` and their problem-specific dimensions (12 for Protein and 30 for UAV).
>
> We will include both these details as well as other detailed experimental settings to make them clear in our revised version. Thank you very much.
>
> [1] MetaBox-v2: A unified benchmark platform for meta-black-box optimization. NeurIPS 2025.
>
> ---
>
> **We hope that our response has addressed your concerns, but if we missed anything please let us know.**

---

> > ### Author Rebuttal · Reviewer_duoh · 2026-04-03
> >
> > Thanks. All my questions have been addressed.

---

> > > ### Author Response · Authors · 2026-04-04
> > >
> > > Dear Reviewer duoh,
> > >
> > > Thank you for your response! We deeply appreciate the time and effort you took to engage with our work.
> > >
> > > We are glad to see that our rebuttal has successfully addressed your questions and concerns, and that the "*(a) Fully resolved - My concerns have been adequately addressed. If you select this option, please consider adjusting your score accordingly*" option was kindly selected. We sincerely hope you might consider reflecting this positive resolution in your overall score, and we would be very grateful for your stronger support.
> > >
> > > Thank you again for your constructive feedback throughout this process, which has genuinely helped strengthen our paper.
> > >
> > > Best regards,
> > >
> > > Authors

---

### Official Review · Reviewer_iyMS · 2026-03-09

**Soundness:** 4
**Presentation:** 3
**Significance:** 4
**Originality:** 4
**Overall Recommendation:** 4
**Confidence:** 3

**Summary:**

The paper introduces Hierarchical Active Task Selection (HATS) to address the "Diversity-Quality Gap" in training Meta-Black-Box Optimization (MetaBBO) agents. HATS uses a bi-level strategy: a robust multi-armed bandit (MAB) allocates training budgets across topological function classes, and an active sampler prioritizes specific instance parameters. This sampling is driven by a regret-based utility metric defined against a heuristic baseline optimizer.

**Compliance With Llm Reviewing Policy:**

Affirmed.

**Final Justification:**

The rebuttal effectively resolved my main concerns by providing ablation studies on teacher bias and clarifying the offline computational overhead. These clarifications demonstrate the framework's robustness, so I decided to raise the score to a 4.

**Key Questions For Authors:**

1) Can you explicitly quantify the total computational overhead required to generate the baseline scores for the utility metric throughout the entire training lifecycle?
2) How sensitive is the emergent curriculum and final agent performance to the choice of the teacher baseline? Have you conducted ablation studies using a weaker heuristic or alternative baseline instead of MadDE?
3) How does the Robust MAB mechanism recover in scenarios where the MetaBBO agent temporarily surpasses the baseline across all available function classes, resulting in uniformly negative utility scores?

If you can address these questions and clarify my concerns, I will raise my score.

**Limitations:**

The framework implicitly assumes that exploring the continuous parameter space yields relatively smooth gradients in problem difficulty. However, affine transformations can trigger discontinuous spikes in landscape hardness. This could destabilize the UAR sampling distribution, causing the replay buffer to over-index on numerically unstable or pathological instances rather than practically solvable ones.

**Strengths And Weaknesses:**

Strengths:
1) Strong Motivational Premise: The empirical demonstration of the "Diversity-Quality Gap", showing that latent feature diversity does not naturally correlate with training efficiency. This is a compelling observation that sets up the problem well.
2) Logical Abstraction: Structuring the BBO instance space hierarchically into global topology and local geometric distortions provides a clean, structured framework for dynamic curriculum generation.
3) Solid Empirical Results: The experimental evaluation across diverse benchmarks (BBOB, CEC) and real-world tasks (Protein Docking, UAV) effectively demonstrates the practical value and versatility of the HATS framework in accelerating convergence and improving final generalization.

Weaknesses:
1) Teacher Bias and Curriculum Bottlenecks: The utility metric is directly anchored to the performance of a baseline algorithm, such as MadDE. If the chosen baseline inherently struggles with a specific multi-modal landscape, the utility gap collapses, potentially causing the MAB and UAR to ignore these instances. Consequently, the learned MetaBBO agent might be constrained by the blind spots of the heuristic baseline.
2) Unacknowledged Computational Overhead: The framework requires evaluating the baseline optimizer on instances to compute the proxy utility score. This introduces a hidden cost in terms of function evaluations, a core metric BBO seeks to minimize. This overhead could be more explicitly discussed in the methodology and experiments.
3) Baseline Comparisons: Comparing the framework primarily against static datasets and basic Domain Randomization leaves room for improvement. The evaluation would be stronger with comparisons against other active sampling or Bayesian optimization-driven data generation baselines.

---

> ### Author Rebuttal · Authors · 2026-03-31
>
> Thank you for your insightful comments. Please find our responses below. The corresponding experimental results are available at https://anonymous.4open.science/api/repo/HATS-CA36/file/iyMS.pdf.
>
> ## W1\&Q2 Teacher bias and curriculum bottleneck
>
> Thank you for this insightful question. Following your suggestion, we conducted an ablation study with alternative teachers (Tab. R1). Teacher choice matters, but not arbitrarily: in most cases, the final HATS agent matches or even surpasses its teacher, indicating that HATS is not strictly bounded by teacher blind spots.
>
> This also addresses concerns in W1. A teacher's blind spot does not automatically make a region invisible to HATS. The utility gap collapses only when both the teacher and the current agent perform similarly poorly on the same instances; when the agent is still worse, the gap remains informative, and those instances are still prioritized.
>
> In addition, HATS is not purely greedy under teacher-defined utility. Class-level MAB preserves exploration across function classes, and UAR prevents low-scored regions from being dropped. Therefore, teacher bias can influence preference, but it does not create a hard curriculum bottleneck. This aligns with Fig. 7, where the learned curriculum remains relatively balanced during training.
>
> We will add these discussions in the revision and leave teacher ensembles/adaptive teacher selection as important future directions. Thank you.
>
> ## W2\&Q1 Quantification of training evaluation overhead
>
> Thank you for your comment. Quantitatively, generating utility scores requires one extra teacher rollout on top of the agent rollout, i.e., roughly doubling the number of function evaluations during training.
>
> However, we would like to clarify that this overhead arises **only during offline MetaBBO training on predefined/generated synthetic functions with oracle access**, which are inexpensive and irrelevant to the evaluation budget on the target test problems. At test time, the learned agent is **frozen**, no teacher is invoked, and all methods are compared under the same function-evaluation budget on the target problems. Therefore, the extra evaluations in HATS should be understood as a **low-cost training-time expense**, rather than an additional query cost on actual test-time BBO tasks.
>
> ## W3 Baselines
>
> Thank you for the helpful suggestion. This paper studies active data generation within MetaBBO training. In this area, most prior work still relies on static training or DR-style augmentation, and recent work such as Diverse-BBO remains essentially *static* rather than actively adjusting the training distribution. To the best of our knowledge, **HATS is a first step toward active data generation in MetaBBO.**
>
> We agree that including stronger active-sampling or BO-driven baselines would further strengthen the evaluation. We attempted such methods while preparing this work, but did not obtain reasonable improvements. More aggressive generation easily produces pathological instances, leading to severe outliers and unstable training. Therefore, HATS intentionally starts from the simpler setting of managing randomly augmented CEC/BBOB candidate instances. Stably generating entirely new instances online with BO-driven or active-generation pipelines remains an important future direction. We will revise the manuscript to include these discussions. Thank you again for your valuable comments.
>
> ## Q3 Robust MAB mechanism when agent surpasses baselines
>
> Thank you for your insightful question. We ran an illustrative experiment for this case: we resumed from the 5-epoch checkpoint and replaced the HATS teacher with random search, making the average batch log-gap uniformly negative. As shown in Fig. R2, performance still improves, and MAB entropy does not collapse. The reason is twofold:
>
> 1. The teacher is only a relative baseline that shapes sampling bias, not a hard learning ceiling.
>
> 2. Robust MAB is updated with clipped rank-based utility loss rather than raw instance-level log-gap.
>
> Thus, even with a negative raw log-gap, the class distribution remains stable, and HATS continues to train effectively. We will add this discussion to the revision. Thank you very much.
>
> ## L1 Replay buffer may be stuck
>
> Thank you for the comment. However, HATS does not assume a smooth difficulty landscape: UAR does not use local perturbations or gradient-like parameter search, but actively selects among randomly generated instances. Moreover, replay is not biased toward the hardest/pathological cases; it prioritizes instances with informative teacher-relative log-gaps. As shown in Fig. R3 (256 generated instances at a mid-training checkpoint), utility has no clear correlation with condition number. Together with the $[u_{\min}, u_{\max}]$ filter, replay focuses on currently learnable tasks rather than over-indexing pathological ones.
>
> ---
>
> **We hope our response has addressed your concerns. If we have missed anything, please let us know.**

---

> > ### Author Rebuttal · Reviewer_iyMS · 2026-04-01
> >
> > I appreciate the authors for their detailed response and the addition of new experimental data, particularly the ablation studies using alternative teachers and the scenario evaluating the MAB mechanism with a random search baseline. The clarifications regarding the training-time computational overhead, along with the empirical evidence demonstrating the stability of the replay buffer and the curriculum, effectively address my concerns about potential teacher bias and curriculum bottlenecks. The authors have fully addressed my primary reservations, demonstrating the framework's effectiveness and robustness, and therefore I have decided to raise my score to 4.

---

> > > ### Author Response · Authors · 2026-04-04
> > >
> > > Thanks for your feedback! We are glad to hear that your concerns have been addressed. We sincerely appreciate the time and effort you have dedicated to reviewing our paper and providing thoughtful and valuable comments.
> > >
> > > Best regards
> > >
> > > Authors

---

### Official Review · Reviewer_QL7R · 2026-03-10

**Soundness:** 2
**Presentation:** 2
**Significance:** 3
**Originality:** 4
**Overall Recommendation:** 4
**Confidence:** 2

**Summary:**

This paper studies data generation for Meta-Black-Box Optimization (MetaBBO). The authors argue that existing training-instance construction methods rely too heavily on landscape diversity, while diversity alone may not reflect how useful a task is for improving the current meta-optimizer. To support this claim, the paper presents an empirical analysis of a “Diversity-Quality Gap,” suggesting that feature diversity correlates poorly with training efficiency. Building on this observation, the paper proposes HATS (Hierarchical Active Task Selection), a utility-aware curriculum generation framework that organizes the instance space hierarchically into function classes and distortion parameters, and then performs bi-level active selection: a robust multi-armed bandit allocates training budget across function classes, while an active sampler prioritizes parameter settings based on regret-based utility relative to a baseline solver.

**Compliance With Llm Reviewing Policy:**

Affirmed.

**Final Justification:**

I believe this work is valuable to the MetaBBO community. However, I still have some remaining doubts, mainly regarding the logical closure of the paper. Therefore, I have raised my score from 3 to 4, but my confidence in the assessment has decreased from 4 to 2. I leave it to the AC to make the final judgment.

**Key Questions For Authors:**

- Q1. How strong is the connection between the paper’s “Diversity-Quality Gap” diagnosis and the actual utility optimized by HATS?

The paper motivates the method by showing that latent diversity is weakly correlated with leave-one-out quality, yet HATS itself is trained using a regret-based utility defined relative to a baseline solver rather than this leave-one-out notion. Could the authors clarify why this proxy is the right one, and provide stronger evidence that optimizing it indeed targets the same notion of training usefulness highlighted in Figure 2?

- Q2. How sensitive is HATS to the choice of the baseline/teacher solver used to define utility?

Since the utility is explicitly defined through the performance gap toward a baseline solver, the induced curriculum may depend strongly on this choice. In the current implementation, MadDE is used as the baseline. How would the curriculum and final generalization performance change if a different teacher were used, or if the teacher were substantially weaker/stronger?

- Q3. How general is the proposed framework beyond the SYMBOL-centered evaluation?

Although the paper presents HATS as a general data-generation framework for MetaBBO, most of the main experimental evidence is built on SYMBOL, while the support on ExPT is relatively limited. Could the authors clarify to what extent HATS is expected to be backbone-agnostic, and whether more evidence can be provided that the gains are not specific to the SYMBOL training setup?

- Q4. Does HATS remain favorable under comparable wall-clock cost, not only in terms of training episodes?

The paper shows faster convergence in terms of episodes, but HATS also introduces additional machinery, including hierarchical scheduling, replay, and utility estimation. Could the authors clarify whether the method remains advantageous under a comparable wall-clock budget, and how much overhead is introduced by the curriculum-generation pipeline itself?

Overall, the question raised by the paper is worthwhile and the motivation is potentially interesting. However, I believe the current version still contains a noticeable gap in its core reasoning chain. In particular, the empirical diagnosis in the early part of the paper is not fully carried through by the subsequent method design and experimental validation. As a result, while the paper starts from an interesting premise, it does not yet deliver a fully self-consistent and convincing technical story. Therefore, I can only give a weak reject at this stage. That said, if the authors can convincingly address the above concerns, I would be open to raising my score.

**Limitations:**

yes

**Strengths And Weaknesses:**

## Strengths

- Meaningful problem setting. The paper highlights an important issue in MetaBBO, namely that training-instance diversity may not be a good proxy for training usefulness. This is a valuable perspective beyond simply proposing another optimizer variant.

- Coherent framework design. HATS combines hierarchical task decomposition, robust bandit-based class selection, and utility-aware replay into a reasonably well-structured curriculum learning framework.

- Broad empirical evaluation. The paper includes multiple benchmarks, application-style tasks, training-efficiency analysis, curriculum-evolution analysis, transfer experiments, and ablations, which gives the work a fairly solid experimental basis.

## Weaknesses

- Proxy mismatch in the core argument. The paper motivates the method using a “Diversity-Quality Gap,” but the actual algorithm optimizes a regret-based utility proxy relative to a baseline solver rather than the proposed leave-one-out quality notion. The connection between the diagnosis and the optimized objective is therefore not fully closed.

- Strong dependence on the chosen baseline solver. Since the utility signal is defined relative to a teacher/baseline solver, the resulting curriculum may be quite sensitive to this choice. This dependency is important but insufficiently analyzed.

- Limited evidence for generality. Although the method is presented as a general framework, the main experimental evidence is still centered on a single backbone, with only limited support for broader backbone-agnostic applicability.

- Practical efficiency remains unclear. The paper shows improvement in training episodes, but it is less clear whether the method remains favorable under comparable wall-clock cost given the added scheduling and replay machinery.

---

> ### Author Rebuttal · Authors · 2026-03-31
>
> Thanks for your valuable comments. Below please find our responses. Corresponding experimental results can be found at https://anonymous.4open.science/api/repo/HATS-CA36/file/QL7R.pdf.
>
> ## W1&Q1 LOO quality and proxy utility
>
> Thanks for your insightful question! The Leave-One-Out (LOO) quality in Fig. 2 is used as a *diagnostic, post-hoc* notion of data usefulness: it measures the marginal contribution of an instance to the final generalization performance, which need retraining the model and evaluating each individual model on a held-out test set. It is suitable for revealing the Diversity-Quality Gap, but **not suitable** as the online utility in HATS due to the heavy LOO retraining and inaccessible evaluations. Moreover, HATS needs a state-dependent signal for the current agent, i.e., whether a task is useful **at the current stage**, rather than whether it directly influences the final model after full training.
>
> Therefore, we use the baseline-relative log-gap utility as an online proxy to show learning potential. Its goal is not to reproduce LOO numerically, but to identify tasks that are **not yet mastered but still learnable** for the current agent.
>
> To understand whether a correct online curriculum can lead to better future performance, we conduct illustrative experiments in Tab. R1 and Fig. R1, showing that 1) training the current agent with appropriate tasks leads to better future performance; 2) HATS steadily schedules tasks with higher LOO quality. These results provide evidence that the utility can be a strong proxy for the LOO quality.
>
> ## W2&Q2 Influence of the baseline/teacher
>
> Thanks for the valuable comment.
> Following your suggestion, we apply multiple teachers (CMA-ES, PSO, and MadDE) on multiple MetaBBO backbones (as your suggestion in W3\&Q3). As shown in Tab. R2, HATS is reasonably robust across different teacher choices, while the overall trend of teachers is consistent across backbones: using MadDE as the teacher generally obtains the best average rank.
>
> A possible reason is that the teacher should also be in the learnable zone for the current agent. A much stronger teacher (e.g., CMA-ES) may make too many tasks look overly hard, while a much weaker teacher may make the utility less informative. From the results, MadDE is slightly weaker than CMA-ES/PSO, but it remains closer to the agent’s capability, yielding a better curriculum.
>
> Overall, these results highlight that teacher selection is important. We thank the reviewer for pointing this out; we will clarify this discussion in the revised version and leave more adaptive teacher design/selection as a future work. Thank you very much.
>
> ## W3&Q3 More backbones
>
> Thanks for your comment. We agree that validating HATS’s effectiveness beyond SYMBOL-centered setting is essential
>
> Following your suggestion and Reviewer duoh's recommendations, we extend the evaluation to additional MetaBBO backbones, including B2Opt [1], POM [2], and LDE [3]. Results in Fig. R2 demonstrate **HATS > DR > Static** across different backbones on both synthetic and real-world tasks. This suggests that the gains of HATS are not specific to the SYMBOL training setup, but arise more generally from improving the training distribution through active task selection.
>
> Additionally, we notice that although HATS uses MadDE as the teacher to define utility during training, the resulting MetaBBO agents can still **match or surpass MadDE** in final performance, indicating the benefit stems from inducing a more effective curriculum rather than mere imitation.
>
> Overall, the results imply that HATS is *backbone-agnostic* and can serve as a general training framework for MetaBBO. Thanks for pointing this out. We will incorporate these results in the revision.
>
> [1] B2opt: Learning to optimize black-box optimization with little budget. AAAI 2025.
>
> [2] Pretrained optimization model for zero-shot black-box optimization. NeurIPS 2024.
>
> [3] Learning adaptive differential evolution algorithm from optimization experiences by policy gradient. IEEE TEvC 2021.
>
> ## W4&Q4 Wall-clock cost
>
> We additionally compare HATS and DR with SYMBOL under wall-clock budget. As shown in Fig. R3, HATS still achieves better test performance earlier in real time.
>
> Using a naive implementation, HATS costs 462s/epoch versus 371s/epoch for DR, due to utility estimation and replay management. However, this overhead can be eliminated. In SYMBOL, teacher trajectories are already available from SYMBOL’s imitation objective, so HATS can directly reuse them for utility computation. Under this implementation, HATS takes 372s/epoch, essentially similar as DR (371s/epoch).
>
> Therefore, the added curriculum does not overturn the practical efficiency advantage of HATS. The main overhead comes from implementation choices, and can be reduced to almost zero when there exists teacher trajectories that can be reused.
>
> ---
>
> **We hope that our response has addressed your concerns, but if we missed anything please let us know.**

---

> > ### Author Rebuttal · Reviewer_QL7R · 2026-04-03
> >
> > Overall, I think the rebuttal is useful and improves the paper. In particular, Q2, Q3, and Q4 are addressed in a fairly convincing way through additional experiments and clearer discussion. The main issue that still remains only partially resolved, in my view, is Q1: the central connection between the paper’s early Diversity-Quality-Gap diagnosis and the actual proxy utility optimized by HATS is now better motivated, but still not fully closed.
> >
> > - Q1: Connection between the “Diversity-Quality Gap” diagnosis and the actual utility optimized by HATS
> >
> > In the paper, the **Diversity-Quality Gap** is diagnosed using the LOO-based quality notion in Figure 2, where function quality is defined by the marginal contribution of an instance to final generalization performance. At the same time, Section 3.2 explicitly states that this LOO objective is too expensive for online use, and therefore HATS replaces it with an instantaneous utility proxy based on the performance gap to a baseline solver.
> >
> > The rebuttal makes this distinction clearer, and I appreciate the explanation that LOO quality is intended as a post-hoc diagnostic, whereas HATS requires a state-dependent online signal that reflects whether a task is useful at the current training stage. I also appreciate the additional evidence suggesting that tasks selected by HATS tend to have higher LOO quality over time.
> >
> > However, **my original concern was not simply why LOO is not used directly, but whether the utility proxy optimized by HATS is sufficiently aligned with the notion of “training usefulness” introduced earlier in the paper.** The rebuttal makes the story more reasonable and adds empirical support, but in my view the connection is still not fully closed. So this concern is alleviated, but not completely removed.
> >
> > -----------------------
> > Thank you again for your response. However, I still remain somewhat uncertain about Q1. I have raised my score, but my confidence in the assessment is reduced.

---

> > > ### Author Response · Authors · 2026-04-04
> > >
> > > Thanks for your timely and constructive response! We are glad to hear that most of your concerns have been addressed and your concern on Q1 have alleviated. We further clarify the intuition and conduct additional experiment based on your advice.
> > >
> > > Conceptually, LOO quality measures whether an instance contains critical landscape features necessary for generalization, thereby indicating training usefulness. Upon this point, the logic linking this to our proxy based on performance gap is straightforward: **If an instance possesses these critical features (high LOO quality), but the agent has not yet mastered them, the agent will naturally perform poorly compared to a feature-agnostic teacher baseline**. This performance discrepancy creates a large performance gap, which is our defined proxy utility. Therefore, the proxy utility serves as a dynamic radar, specifically isolating instances that contain high-value, unmastered features.
> > >
> > > To provide direct evidence for this logic, we conducted an illustrative experiment **aiming to show that the utility can be an effective proxy for local LOO quality**. As shown in https://anonymous.4open.science/api/repo/HATS-CA36/file/loo_utility_vs_quality.pdf, we plotted the *Utility* (Log Performance Gap, X-axis) against the actual *Local LOO Quality* (Marginal test performance gain after 100 training steps, Y-axis) of 32 randomly generated instances. We can find that our utility exhibits a strong positive correlation (Pearson $r=0.734$, Spearman $\rho=0.675$) with the local LOO quality. This clearly shows that the instances our proxy actively selects tend to be the ones delivering the highest marginal training usefulness by achieving a higher LOO performance improvement. Thus, as HATS actively selects and replays instances that possess higher utilities, the learned agent adaptively achieves higher performance gain, resulting in better final performance. We will gladly include this discussion in the revised paper.
> > >
> > > **Thank you again for your constructive feedback to help improve our paper's quality! We will include this discussion in the revised paper. Thank you very much.**

---

### Official Review · Reviewer_wTut · 2026-03-15

**Soundness:** 2
**Presentation:** 1
**Significance:** 2
**Originality:** 2
**Overall Recommendation:** 3
**Confidence:** 3

**Summary:**

This paper studies meta black box optimization, where the authors identify a fundamental gap in the performance of algorithms for these problems. Increasing the diversity of the optimization landscape can help during training, but doing so with an active task selection strategy can be much more beneficial. The authors then propose a hierarchical active task selection framework for meta black box optimization based on evaluating a regret metric to select the tasks to be added to the training mechanism.

**Compliance With Llm Reviewing Policy:**

Affirmed.

**Final Justification:**

The author's response and the discussion with other reviewers helped mitigate some of my concerns.

**Key Questions For Authors:**

I hope the authors take this feedback and revise their manuscript so all terms are well defined.

**Limitations:**

In my opinion the paper needs a rewrite.

**Strengths And Weaknesses:**

The paper studies an important problem in meta learning, the curriculum of task selection for generating a meta optimizer is important. Developing insights on how different task schedules can help optimization is an important problem. Unfortunately I think the paper is extremely hard to understand. It is impossible to know what is the output of $\pi_{\phi}$ is it actions? what is the domain of these actions. What is the domain of $x$?

Similarly throughout the rest of the document the writeup is plagued by inaccuracies, and undefined terms. For example, what is $g_c$ above section 4? What are the $f$ in section 2.3?

---

> ### Author Rebuttal · Authors · 2026-03-31
>
> Thank you for your comment.
> However, we respectfully disagree with this point. The concern appears to stem from a misreading of the notation, as all referenced symbols are explicitly and unambiguously defined in the paper.
>
>
> - $x$ is defined in Section 2.1 as the query point in the search space $\Omega \subseteq \mathbb{R}^D$, with objective $f:\Omega \to \mathbb{R}$ and response $y=f(x)$.
>
> - $\pi_\phi$ is defined immediately below as the **parameterized policy** controlling optimizer $A$, where $\phi$ denotes the meta-parameters. Running $A(\pi_\phi, f)$ on a problem instance $f$ generates the trajectory $\tau_f=  \\{  (x_t,y_t)  \\}  _{t=1}^T$.
>
> - $g_c$ is explicitly defined in Section 3.3 through $f(x;c;\theta)=g_c(T(x;\theta))$, where $g_c$ denotes the **canonical base function** and $T$ denotes the geometric transformation.
>
> - The symbol $f$ does **not** appear in Section 2.3. It is also not a new symbol; rather, it is the **problem instance / objective function** already introduced in Section 2.1.
>
> Given these explicit definitions, we believe the notation is clear and internally consistent. We would appreciate reconsideration of this concern and sincerely look forward to your further feedback.
>
> **We hope our response has addressed your concerns. If we have missed anything, please let us know.**

---

> > ### Author Rebuttal · Reviewer_wTut · 2026-04-05
> >
> > The authors response has assuaged some of my concerns. I still think the work is very hard to read.

---

### Decision · Program_Chairs · 2026-04-30

**Decision:**

Reject

**Comment:**

This paper introduces Hierarchical Active Task Selection (HATS), an automated curriculum framework designed to improve the data generation process for training Meta-Black-Box Optimization (MetaBBO) agents. The framework utilizes a bi-level strategy to dynamically allocate training budgets across function classes and prioritize specific instance parameters using a regret-based utility metric.

This paper tackles a relevant problem and has solid potential. While reviewers agreed that the empirical identification of the "Diversity-Quality Gap" is a compelling premise and that the framework's methodology is logically structured, they also raised important concerns about the paper's theoretical tightness, evaluation baselines, and overall presentation:

1. The experimental evaluation primarily compares HATS against static datasets and standard Domain Randomization. The paper lacks comparisons against other active sampling or Bayesian Optimization-driven data generation baselines. The authors acknowledged this limitation in the rebuttal, noting that such methods easily produced pathological instances in their testing, which leaves the empirical validation against true active-generation competitors incomplete.
2. There remains a logical gap between the paper's motivational diagnostic metric (Leave-One-Out quality) and the actual operational proxy metric optimized by the algorithm (regret relative to a baseline). While the authors provided new empirical data showing a strong correlation between the two during the rebuttal, reviewers felt that the theoretical connection bridging this core argument is still not fully closed.
3. Multiple reviewers raised concerns regarding the manuscript's exposition. Even after the rebuttal clarified specific notational misunderstandings, reviewers maintained that the manuscript remains hard to read and requires nontrivial revision for clarity.

Therefore, the paper cannot be accepted to the conference at this time. The core ideas are original and the expanded experiments provided during the rebuttal demonstrate clear promise. I encourage the authors to revise the manuscript to address these theoretical, empirical, and presentational issues and resubmit to an appropriate future venue.

**Additional Remarks:**

As you revise this work for future submission, consider expanding the literature review to capture missing context discussed during the review period. Specifically, the next iteration of this paper would benefit from:
* Comparisons or direct references to continuous Automatic Curriculum Learning (ACL) papers (e.g., GoalGAN, ALP-GMM), which tackle similar continuous parameter sampling challenges.
* Integration of the recent MetaBBO architectures that were successfully evaluated during the rebuttal (e.g., POM, B2opt) into the main text to ensure the baselines reflect the current state-of-the-art.